# The Role of Very Low Calorie Ketogenic Diet in Sympathetic Activation through Cortisol Secretion in Male Obese Population

**DOI:** 10.3390/jcm10184230

**Published:** 2021-09-18

**Authors:** Rita Polito, Giovanni Messina, Anna Valenzano, Alessia Scarinci, Ines Villano, Marcellino Monda, Giuseppe Cibelli, Chiara Porro, Daniela Pisanelli, Vincenzo Monda, Antonietta Messina

**Affiliations:** 1Department of Advanced Medical and Surgical Sciences, Università degli Studi della Campania “Luigi Vanvitelli”, 80138 Naples, Italy; rita.polito@unicampania.it; 2Department of Clinical and Experimental Medicine, University of Foggia, 71100 Foggia, Italy; anna.valenzano@unifg.it (A.V.); giuseppe.cibelli@unifg.it (G.C.); chiara.porro@unifg.it (C.P.); Daniela.pisanelli@unifg.it (D.P.); 3Department of Education, Psychology, Communication, University of Bari, 71121 Bari, Italy; alessia.scarinci@uniba.it; 4Department of Experimental Medicine, Section of Human Physiology and Unit of Dietetics and Sports Medicine, Università degli Studi della Campania “Luigi Vanvitelli”, 80138 Naples, Italy; ines.villano@unicampania.it (I.V.); marcellino.monda@unicampania.it (M.M.); vincenzo.monda@unicampania.it (V.M.); antonietta.messina@unicampania.it (A.M.)

**Keywords:** cortisol, galvanic skin response (GSR), very low-calorie ketogenic diet (VLCKD), adipose tissue, obesity, sympathetic nervous system, hypothalamus-pituitary-adrenal (HPA) axis

## Abstract

Adipose tissue is considered an endocrine organ, and its excess compromises the immune response and metabolism of hormones and nutrients. Furthermore, the accumulation of visceral fat helps to increase the synthesis of cortisol. The hypothalamus-pituitary-adrenal (HPA) axis is a neuroendocrine system involved in maintaining homeostasis in humans under physiological conditions and stress, and cortisol is the main hormone of the HPA axis. It is known that a stress-induced diet and cortisol reactivity to acute stress factors may be related to dietary behavior. In obesity, to reduce visceral adipose tissue, caloric restriction is a valid strategy. In light of this fact, the aim of this study was to assess the effects of a commercial dietary ketosis program for weight loss on the sympathetic nervous system and HPA axis, through evaluation of salivary cortisol and GSR levels. Thirty obese subjects were recruited and assessed before and after 8 weeks of Very Low Calorie Ketogenic Diet (VLCKD) intervention to evaluate body composition and biochemical parameters. Salivary cortisol levels and GSR significantly decreased after dietary treatment; in addition, body composition and biochemical features were ameliorated. The VLCKD had a short-term positive effect on the SNS and HPA axes regulating salivary cortisol levels. Finally, the effects of the VLCKD on the SNS and HPA axis may lead to more individualized treatment strategies that integrate obesity and stress and support the usefulness of such therapeutic interventions in promoting the reduction of the individual disease burden.

## 1. Introduction

The center of stress management consists of the sympathetic nervous system (SNS) and the hypothalamic-pituitary-adrenal axis (HPA). These systems coordinate the response of the various physiological systems, leading the body to equilibrium. There is robust evidence from animal and a human study that stress causes coincidental activation of nerve and system cells and also the unharnessing of many biologically active compounds, as well as catecholamines and glucocorticoids [1,2]. Dysregulation of one of those stress systems will result in an altered nervous response [3,4]. Activation of the SNS, thanks to stress, is measurable by numerous parameters, such as galvanic skin responses (GRS), a modification in electrical physical phenomenon between two points on the respondent’s skin [5]. Additionally, the HPA axis may be a system concerned with maintaining equilibrium in mammalian organisms under physiological conditions and stress, and hydrocortisone is the main hormone of the HPA axis in humans [6,7,8]. The evaluation of secretion of hydrocortisone can be a valid and easy parameter for HPA axis assessment.

Cortisol receptors, present in most cells in the body, receive and use the hormone in different ways. For example, when the body is on alert, cortisol can alter or disrupt the functions that stand in the way [9]. These can include the digestive or reproductive systems, the immune system, or even growth processes. In addition, it is known that a stress-induced diet and cortisol reactivity to acute stress factors may be related to dietary behavior. In addition, in obesity there is a high level of cortisol, in fact there is a close relationship between adiposity and increased levels of cortisol [10,11,12]. With hyperbolic stress, internal secretion of corticoid adrenal cortical steroid plays a role in the development of fat. However, it appears that not all people respond to stress with the same means. This raises the question of whether there is an associated degree of interindividual variation within the biological response to stress. Adrenal cortical steroid causes a distribution of white animal tissue within the abdominal region and will increase appetite, with a preference for high energy density foods (“comfort food”) [13]. Patients that the United Nations Agency Area Unit inveterately exposed to high levels of glucocorticoids, such as in Cushing syndrome or with exploitation of high doses of exogenous glucocorticoids, develop abdominal fat, metabolic syndrome, and eventually disorder. Interestingly, in our contemporary society, the fat pandemic coincided with a rise in factors that hyperbolize the assembly of adrenal cortical steroids, such as chronic stress, the consumption of high glycemic index foods, and a reduced quantity of sleep [14]. This implies a vicious circle, within which the hyperbolic action of glucocorticoids, fat, and stress increase and amplify one another. This hypothesis is supported by recent studies showing vital correlations between fat and long-term adrenal cortical steroid levels, measured in scalp hair, both in adults [15] and children [16,17]. Animal tissue is considered a type of endocrine organ and its excess compromises immune reactions and the metabolism of hormones and nutrients. In addition, the buildup of visceral fat helps to extend the synthesis of adrenal cortical steroids [17]. Obesity and hunger have different effects on normal physiology and are associated with adaptive changes in hormonal secretion. Many data in the literature also report that a balanced diet induces a change in body composition and body weight of obese subjects, as well as acting on the activity of cortisol [12,13,14,15,16,17]. In this scenario, it is known that many obesity-related diseases can be countered with proper nutrition and/or calorie restriction, by reducing or slowing down the onset of numerous inflammatory diseases and having countless beneficial effects by reducing oxidative stress [18,19]. For these reasons, nutritional interventions, such as calorie-restricted diets, can be an effective therapeutic approach to promote weight loss in obese patients and decrease salivary cortisol and its effects [20]. The very low carbohydrate ketogenic diet (VLCKD) induces rapid weight loss and is also able to improve hyperlipidemia and certain cardiovascular risk factors; therefore, the VLCKD has proven to be a valuable tool for combating obesity, in an average time span of 3 to 6 months [21,22]. Considering this evidence, this study aimed to assess the effects of a commercial dietary ketosis program for weight loss on the sympathetic nervous system and hypothalamus-pituitary-adrenal axis (HPA) through evaluation of salivary cortisol and GSR levels in obese subjects, to demonstrate the role of cortisol in the establishment of obesity, and not only as a sympathetic nervous system expression but also as a metabolic factor modifiable by diet.

## 2. Materials and Methods

### 2.1. Subjects, and Anthropometric and Biochemical Measurements

Thirty male subjects, affected by obesity and aged between 50 and 60 years (±8 years), were enrolled by the Laboratory of Physiology, Department of Clinical and Experimental Medicine, University of Foggia. The study was approved by the local Ethics Committee 22 May 2018, n◦440/DS, and conducted according to the ethical principles of the Declaration of Helsinki. Written informed consent was obtained from all participants. 

The sample chosen was completely male because, given the age range of our population (50–60 years), we preferred to consider only a male population to minimize hormonal interference due to menopause, which could have affected a group of women of the same age. In our population, we evaluated body composition by measuring height, weight, body mass index (BMI), and visceral adipose tissue (VAT) using dual-energy X-ray absorptiometry (DEXA). Serum samples were collected after a 12-h overnight fasting period. Serum aliquots were stored at −80 °C. As previously reported, for all participants, glucose, total cholesterol, low-density lipoprotein (LDL), high-density lipoprotein (HDL), triglycerides, LDH, aspartate transaminase (AST), alanine transaminase (ALT), IL-10, TNF-a, adiponectin, and Orexin-A were measured [21,22].

### 2.2. Study Protocol 

All participants underwent a general medical examination [9]. To verify if the patients followed the diet correctly, every day ketone was measured in capillary blood (GD40 Delta test strips, TaiDoc Technology Co.; New Taipei City, Taiwan). The levels b-hydroxybutyrate >0.5 mmol/L determined the nutritional ketosis. Fasting (12 h) blood samples were collected at 8:00 a.m. [21,22].

### 2.3. Diet Protocol

Obese subjects followed a VLCKD, as per an advertised weight loss program (Lignaform, Therascience), consisting of <50 g/d sugar from vegetables, 43% fat, 43% protein, 14% carbohydrates, and 700–900 kcal. The amount of protein ranged between 1.0 and 1.2 g per kilo of ideal weight. Although, the dietary intervention profile consisted of three totally different stages, for the needs of this study, only active ketogenic phases of the primary stage were considered. This stage had an 8-week length, to permit participants to realize some of their weight loss target. In this period, vitamins, minerals, and polyunsaturated fatty acid were provided in accordance with international recommendations.

### 2.4. Hormonal Secretion Assay

Salivary samples were collected between 09:00 and 11:00 before the VLCKD intervention and after eight weeks of VLCKD intervention using of cotton swabs (Salivette, Sarstedt, Rommelsdorf, Germany). Participants were asked to position the cotton swab in their mouth for a minimum of two minutes and then insert it into a special plastic tube. Samples were sent as soon as possible to the laboratory and kept at −20 °C until the corticoid assay. The secretion samples were centrifuged at 1500× *g* for 15 min at 4 °C. To gauge the absence of blood contamination, a secretion blood contamination kit was used (Salimetrics LLC, State faculty, PA, USA). Corticoid concentrations were measured using commercial kits (Salimetrics LLC, State faculty, La Place Court, Carlsbad, CA, USA), as per the manufacturers’ directions. All samples were tested in triplicate and analyzed in duplicate.

### 2.5. Galvanic Skin Response

Within five minutes following spittle collection, electrodermal response information was recorded. The electrodermal response parameters were measured with a SenseWear professional Armband™ (version 3.0, BodyMedia, Pittsburgh, PA, USA), which was worn on the appropriate arm over the striated muscle at the center between the appendage and olecranon process, as suggested by the manufacturer [5].

### 2.6. Applied Mathematics Analysis

We performed applied mathematical analyses using StatView software 5.0.1.0 (SAS Institute, Cary, NC, USA). All information is given as means ± SE; a two-tailed paired t test was used to check for applied mathematical significance of the outcome variables. A *p*-value ≤ 0.05 was used for statistical significance. Secretion of corticoid, adiponectin humor concentrations, and total cholesterol were related to Pearson’s tests, as per the information distribution. A *p*-value < 0.05 was considered statistically significant.

## 3. Results

### 3.1. Anthropometric and Biochemical Characteristics of VLCKD Obese Patients

In the VLCKD patients after diet intervention, there was a large change in anthropometric and biochemical parameters. Many parameters such as weight and BMI were statistically reduced in VLCKD obese subjects after the diet. Furthermore, *t*-test analysis showed that glycemic, lipid profile, and inflammatory status were strongly ameliorated in these subjects after the diet. In Table 1 we report the results, with a *p*-value < 0.05. 

### 3.2. Physiological Responses to VLCKD

The VLCKD induced a significant decrease in both autonomic and hormonal variables. The analysis of variance showed significant differences between the experimental conditions for GSR (4.2 μS vs. 2.11 μS, *p* < 0.05) and cortisol (1.31 μmol/L vs. 1.01 μmol/L *p* < 0.01). Salivary cortisol levels in the VLCKD obese patients were strongly decreased after the VLCKD intervention (Figure 1). In Figure 2 reports the GSR modulation. Interestingly, using Pearson’s test correlation, we observed correlations between salivary cortisol, biochemical, and anthropometric parameters in the VLCKD obese subjects. Salivary cortisol negatively correlated with BMI and total cholesterol and positively correlated with adiponectin serum levels, expressed as μ variation in Figure 3A–C. 

## 4. Discussion 

In obesity, there are several aspects to contemplate such as secretion and metabolic processes, as well as aerophilous stress, inferior chronic inflammation, and hormone resistance [23]. It is fascinating to note that secretion alterations play a vital role within the development of obesity.

In addition, many studies on obesity have shown alterations in corticosteroid metabolism and in its secretion and action [24]. In our study, we investigated the consequences of a calorie restriction program for weight loss on the sympathetic system and hypothalamus-pituitary-adrenal axis (HPA) through the analysis of the secretion of corticosteroid and reaction levels.

We found a distinct reduction in corticosteroid concentrations within the saliva of the obese subjects with VLCKD intervention. Several pieces of information within the literature suggest that the deregulation of the neural structure of the adrenal–pituitary axis contributes to its disorder, increasing the secretion of corticosteroid, which may be a risk factor for numerous metabolic disorders [25]. Corticosteroid is a vital regulator of endocrine operation, metabolism, and differentiation of adipocytes, as well as a contributor to adipogenesis and increased visceral fat reserves [26]. In addition, this hormone plays a major role within the hormone signaling pathways, because it alters the hormone sensitivity in several tissues, thereby reducing aldohexose uptake and contributing to hormone resistance [27]. Corticosteroid metabolism is regulated by two groups of enzymes, 11βdehydrogenase [9]. Increased expression of the protein is related to the pathological process of central obesity, metabolic syndrome, and the dysregulation of aldohexose and lipoid metabolism [9]. Studies have shown an association between increased expression and activity of this protein in visceral white animal tissue and reduced plasma concentrations of adiponectin, which might indirectly contribute to increased production and release of pro-inflammatory cytokines (as in [3,28,29,30,31,32]), supporting of the correlation of statistics that we tend to find between corticosteroid and adiponectin. The role of glucocorticoids in the regulation of animal tissue operation is very advanced. These substances result in the differentiation of preadipocytes into mature adipocytes, similarly to fat tissue lipolysis under certain conditions [33,34,35]. Chronic exposure to glucocorticoids favors the enlargement of animal tissue, which compromises the action of the hormone, leading to symptoms and dyslipidemia.

In a lean state, the corticosteroid is lower, and its production will increase in response to stress, as well as diet interventions. In an obese condition this is totally different. In fact, the consequences of stress on weight gain manifest in several ways because of the various properties of glucocorticoids. High levels of a corticosteroid will, for example, increase appetite, with a preference for “comforting food”, and cause the distribution of white animal tissue within the abdominal region, which might eventually result in abdominal weight gain [14]. Curiously, it has been determined that glucocorticoids will scale back the sensitivity of the adrenergic stimulation of brown fat [14]. Additionally, the administration of exogenous glucocorticoids will increase the intrahepatic conversion of ketosteroids to corticosteroids; therefore, undoubtedly contributing to the vicious circle [15,36]. This relationship between chronic stress-associated weight mediate by an increased action of glucocorticoids could also be bigger in some folks because of exposure to factors that improve stress response. Biological factors, like the transport of glucocorticoid-sensitive GR factor variants, or a diurnal cortico-steroid rhythm interrupted by shrunken sleep and/or shift work, will doubtless result in bigger effects of glucocorticoids and therefore build some folks a lot of liable to weight gain and obesity [36,37]. On the opposite hand, obesity itself also can result in a rise in chronic stress a variable degree, depending on certain individual characteristics. People that experience, for example, weight stigma are identified not only as experiencing a lot of stress, but additionally to have higher semi-permanent corticosteroid levels [14,36,37]. Additionally, obese patients are a lot more likely to suffer from mental and physical disorders that successively will result in chronic stress and/or higher corticosteroid levels [23,38]. This might even be ex-aggregated by using certain medications indicated for comorbidities associated with obesity, such as corticosteroids for arthritis or respiratory illness. Additionally, the same applies to alternative environmental and activity factors, such as the intake of foods with a high glycemic index, excessive use of alcohol, and chronic pain, which might itself result in a rise in corticosteroid levels and a rise in weight [14].

In this scenario, a diet reducing carbohydrates, such as the VLCKD can reduce cortisol levels and have many beneficial effects, not only on weight loss but also for re-modulating metabolic, hormonal, and psychological aspects [21,22,24].

In addition, the treatment of obesity with very-low-calorie diets causes a decrease in serum cortisol, explained by the decrease in cortisol-binding proteins. The increased cortisol secretion observed in patients with abdominal obesity can contribute to metabolic syndrome (insulin resistance, glucose intolerance, dyslipidemia, and hypertension) [24]. In addition, the role of stress in obesity is also reported in this study. Several studies have shown that the high responsiveness of cortisol to stress is related to increased food intake in healthy-weight individuals [38].

Our data provide further support for a possible effect of cortisol reactivity on food intake in people with obesity, but not in healthy weight controls. Therefore, the physiological mechanism of stress-induced nutrition is a complex interaction between many different hormones, whose secretion and activity are affected by glucocorticoids [3,38,39,40,41]. 

In addition, we also found a reduction in GSR, which confirms the role of stress and the reactivity of cortisol in food intake, confirming the association between the SNS and the HPA axis [3,38,39]. 

## 5. Conclusions

A VLCKD has a short-term positive effect on the SNS and HPA axes. In addition, there is a strong relation between obesity and stress, and cortisol has a key role. For these reasons, VLCKD has an important role reducing stress and regulating salivary cortisol levels. Finally, the effects of VLCKD on the SNS and HPA axis may lead to more individualized treatment strategies that integrate obesity and stress in support of the reduction of the individual disease burden.

Limitation of the study: this is one of the first studies investigating VLCKD effects on the SNS and HPA axis; however, a limitation to this study is the small number of male patients considered. For this reason, it is our intention to expand the series to include the female population in further studies on this subject.

## Figures and Tables

**Figure 1 jcm-10-04230-f001:**
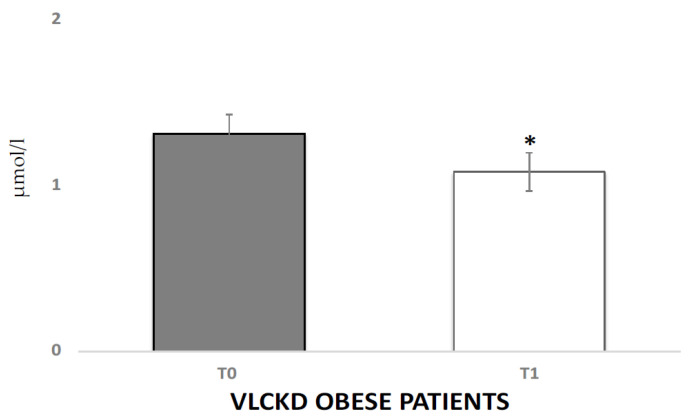
Salivary cortisol levels before and after the VLCKD in obese subjects. μmol/L (micromole/litro); (*) inidcates a significant difference (*p* < 0.05) with respect to the basal value.

**Figure 2 jcm-10-04230-f002:**
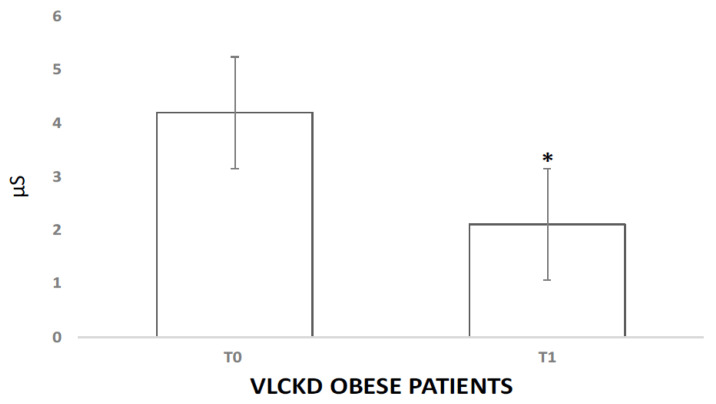
Changes in skin conductance level. μS (micro-Siemens); (*) indicates a significant difference (*p* < 0.05) with respect to the basal value.

**Figure 3 jcm-10-04230-f003:**
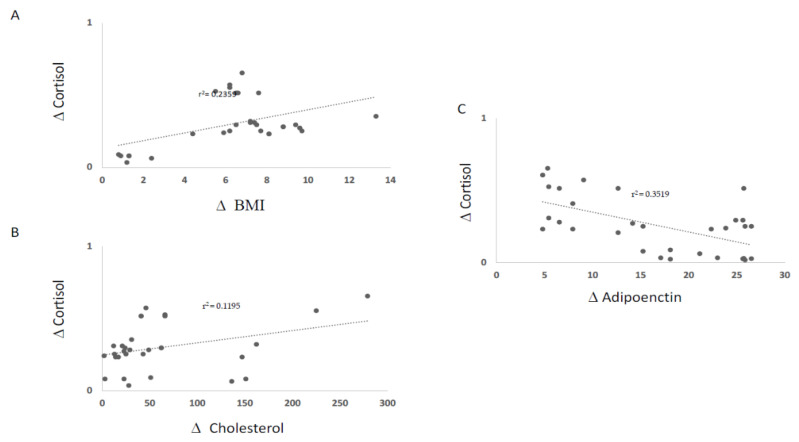
Salivary cortisol concentrations correlated negatively with BMI and total cholesterol (**A**) and (**B**); and positively correlated with adiponectin (**C**). (Δ) indicates the variation between T1 and T0 in the VLCKD obese subjects for each parameter.

**Table 1 jcm-10-04230-t001:** Principal characteristics of the VLCKD obese population before and after the diet.

	VLCKD Obese Subjects		
	T0	T1	*p*-Value
Age	57 ± 8		ns
Height (m)	1.65 ± 0.2		ns
Weight (kg)	90.73 ± 11	83.33 ± 12	<0.05
BMI (kg/m²)	33 ± 4.5	30.5 ± 3.22	<0.05
Total cholesterol (mg/dL)	220.13 ± 50.77	173.91 ± 32.93	<0.05
Triglycerides (mg/dL)	126.54 ± 11	90.25 ± 14	<0.05
AST-GOT (U/L)	22.17 ± 5.98	20.31 ± 4.7	<0.05
ALT-GPT (U/L)	26.01 ± 14.89	24.06 ± 16.27	<0.05
Gamma GT (U/L)	30 ± 8.8	15.31 ± 5.41	<0.05
Azotemia (mg/dL)	36.7 ± 8.43	35.01 ± 5.16	ns
Calcemia (mg/dL)	9.7 ± 0.33	9.52 ± 0.35	ns
Sodium (mmol/L)	139.19 ± 2.48	139.18 ± 2	ns
CRP (mg/mL)	0.89 ± 0.1	0.48 ± 0.07	<0.05
Adiponectin (μg/mL)	9.96 ± 0.7	23.56 ± 2.33	<0.001

T0 (time before Very Low Calorie Ketogenic Diet intervention); T1 (time before Very Low Calorie Ketogenic Diet intervention; BMI (Body Mass Index); AST-GOT (Aspartate aminotransferase); ALT-GPT (Glutamate-pyruvate transaminase); Gamma GT (Gamma glutamyl transferase); CRP (C-reactive protein); ns (not significant).

## Data Availability

Data is contained within the article. Authors can use this data for research purposes only by citing our research article.

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
