# Peer review of "The Role of Very Low Calorie Ketogenic Diet in Sympathetic Activation through Cortisol Secretion in Male Obese Population"

_jcm, 2021, doi:10.3390/jcm10184230_

Round 1

Reviewer 1 Report

I had one major criticism/suggestion, and several very minor suggestions.  The major one is that the only description of the intervention was "a commercial dietary ketosis program for weight loss"   I was curious about what it was?  were they eating regular food?  were they eating "specialized commercial food" what was the fat/carb/protein ratio etc etc  I was also curious about how many patients were in this program.  Did this study only address those who were successful? who maintained ketosis?  If not, were the others studied? How many were there? If it studied only those who achieved success, that is fine, but it should be stated.   The minor points are "picky" suggestions to use more exact language. Specifically

line 62  rather than "improved" say "lead to higher" or "increased"

line 70  you imply obesity and hunger are opposites

line 72 you note a "proper diet" but that implies a  judgement I do not understand, and it is poorly defined

line 85 you note obesity subjects but I think you meant obese subjects

line 95 you say VAT by DEXA but should spell out visceral adipose tissue and dual energy Xray absorption, because it is not clear.

In general though, this paper was very good.

Author Response

In attached our poin-by-oint response.

Reviewer 2 Report

    This is a very interesting study with a good design, but it has a serious problem, the sample.  The authors will have to convince that the small sample of the study is representative.

Below are my comments and suggestions for the authors.

Title

The title should make it clear that the study was conducted in a male population only.

  1. Materials and Methods

2.1. Subjects, anthropometric and biochemical measurements

In this part of the paper, the authors should explain how the sample was selected: why only a male population?  Was there any randomized selection from a larger sample?

2.2. Study Protocol 

I would like to ask you to add a small paragraph here, describing the basic elements of the diet.

2.5. Statistical Analysis

Please add which statistical methods were used other than correlation.

Which of your variables followed a normal distribution, which ones used Pearson's correlation and which ones used Spearman's test?

  1. Results

Line 132-135: Add which statistical method you used, references to be removed.

Table 1: From which statistical method did the p , fill in the table.

Line 142-146: Which correlation method was used and on which variables? It is useful to add the r value in addition to p.

Figure 3: I think they should be replaced with a single table to show the correlations of all variables.

  1. Discussion

Line 165: ".... after VLCKD surgery" what is meant?

Line 203-204: I think the sentence should be deleted, if you choose to keep it add a reference.

 5 Limitations of the Study: I think it should be added.

Author Response

In attached point-by-point response.
